# Exosomes Derived from Non-Classic Sources for Treatment of Post-Traumatic Osteoarthritis and Cartilage Injury of the Knee: In Vivo Review

**DOI:** 10.3390/jcm10092001

**Published:** 2021-05-07

**Authors:** Dan Li, Puneet Gupta, Nicholas A. Sgaglione, Daniel A. Grande

**Affiliations:** 1Orthopedic Research Laboratory, The Feinstein Institute for Medical Research, Northwell Health System, Manhasset, NY 11030, USA; dli5@northwell.edu; 2Department of Orthopaedic Surgery, George Washington University School of Medicine and Health Sciences, Washington, DC 20052, USA; guptap14@gwu.edu; 3Department of Orthopaedic Surgery, Long Island Jewish Medical Center, Northwell Health, New Hyde Park, NY 11040, USA; nsgagli@northwell.edu

**Keywords:** exosomes, osteoarthritis, cartilage injury, regenerative medicine

## Abstract

Osteoarthritis of the knee is one of the most common chronic, debilitating musculoskeletal conditions. Current conservative treatment modalities such as weight loss, non-steroidal anti-inflammatory drugs, and intra-articular steroid injections often only provide temporary pain relief and are unsatisfactory for long-term management. Though end stage osteoarthritis of the knee can be managed with total knee arthroplasty (TKA), finding alternative non-surgical options to delay or prevent the need for TKA are needed due to the increased healthcare costs and expenditures associated with TKA. Exosomes have been of particular interest given recent findings highlighting that stem cells may at least partially mediate some of their effects through the release of extracellular vesicles, such as exosomes. As such, better understanding the biological mechanisms and potential therapeutic effects of these exosomes is necessary. Here, we review in vivo studies that highlight the potential clinical use of exosomes derived from non-classical sources (not bone marrow or adipose derived MSCs derived MSCs) for osteoarthritis of the knee.

## 1. Introduction

Osteoarthritis (OA) is a leading cause of chronic disability and pain with an estimated nearly 9.6% of men and 18% of women over the age of 60 across the world suffering from OA [1,2]. Current conservative management consisting of non-steroid anti-inflammatory drugs (NSAIDs), intra-articular glucocorticoid injections, physical therapy, and more are often unsatisfactory. Currently there are no drugs that directly treat the disease to prevent the pathologic progression to its end stage, to which the only treatment is total knee arthroplasty (TKA). Due to the lack of curative therapy, TKA had its utilization rates more than double from 1999 to 2008 [3]. Most importantly, an estimated nearly more than $27 billion in healthcare costs annually can be attributed to knee OA [4,5,6]. Given such large healthcare expenditures and current inadequate multimodal treatment regimens, developing novel non-surgical or minimally invasive therapeutics is critical.

One growing area of research regarding alternative methods for treating OA has been focused on the use of mesenchymal stem cells (MSCs) due to their ability to proliferate and differentiate into other cell lineages, thereby potentially regenerating cartilage, bone, and other structures [7,8]. MSCs have been shown both in vivo and in vitro to play an important role in protecting against the development of OA or articular cartilage destruction via anti-apoptotic activity, anti-fibrotic activity, promoting chondrogenesis, immunosuppression, regulating metabolic activity, and more [9]. Contrary to conventional thoughts, further investigations with MSCs showed that many of its beneficial effects in OA likely involve paracrine signaling, of which one is via the release of extracellular vesicles (EVs) [10,11]. A subtype of these EVs are exosomes, 30-150 nm diameter particles secreted from cells for intercellular communication that carry proteins, lipids, nucleic acids (including mRNAs and microRNAs (miRNA)), and more [12]. As such, many investigations are currently in progress to better understand the roles and mechanisms by which exosomes function and the potential for them to be used clinically in OA and other pathologies.

There are multiple advantages and properties of exosomes compared to MSCs that improve the likelihood of them being translated clinically and thus the recent rapid rise in the amount of literature about them. For example, given the small size and hypoimmunogenic nature of exosomes, there is a lower likelihood of rejection [13,14]. Additionally, as exosomes are non-viable, lower costs are involved in their maintenance and storage relative to MSCs, as cells need to be maintained in a viable state [13,14]. Furthermore, exosomes have the ability to cross the blood-brain barrier, opening the possibility for any therapies targeting the central nervous system or for future use as drug-delivery vehicles [15]. Such benefits of exosomes over MSCs and other cell-based therapies have further contributed to the rising interest in exosomes.

For OA of the knee, current literature has extensively reviewed the progress made in understanding exosomes derived from bone marrow-derived MSC exosomes (BMSC-exosome) as well as adipose tissue-derived MSC exosomes (ADSC-exosome) (classic MSC-derived exosome sources for knee OA), given their more direct relevance to cartilage destruction and regeneration. However, given the plethora of cell types in the knee joint, early investigations into other non-classic sources of MSC-derived exosomes or exosomes from other tissues have been completed and shown promising results. Here we review the advances made in understanding non-classical sources of exosomes, including but not limited to synovial derived exosomes, infrapatellar fat pad derived exosomes (IPFP-Exos), and even platelet-rich plasma derived exosomes (PRP-Exos).

## 2. Materials and Methods

### Search Strategy

An organized literature search was conducted using the PubMed database. Keywords used for the initial search included “cartilage or osteoarthritis” and “exosomes” with only publications from 15 January 1990 to 15 January 2021 included. The reference lists of these studies were also manually evaluated to identify any other relevant studies. Additional searches of the database using other terms such as “synovial membrane exosomes” were also conducted to ensure an in-depth review. Only animal studies written in English that explored the use of exosomes derived from non-classic sources for knee osteoarthritis or focal cartilage defects were included.

## 3. Results

### 3.1. Literature Analysis

A total of 161 studies were obtained from the initial query search. After evaluating these reports, their reference lists, and using additional search terms, a total of 12 studies met the criteria for analysis described previously (Figure 1) [11,16,17,18,19,20,21,22,23,24,25,26]. The exosomes in these 12 studies were derived from various sources, as listed in Table 1. One study used exosomes derived from platelet-rich plasma, one from infrapatellar fat pads, three from synovial membranes, three from the umbilical cord, three from human embryonic stem cells, and one from amniotic fluid. Six of the studies were conducted using a rat model, four studies used mice, and two studies used rabbits for their investigations.

### 3.2. Non-classic Exosomal Sources

As mentioned previously, the biology, preparation, characterization, and application of exosomes derived from classical MSC sources have been extensively reviewed in prior literature [27,28,29,30,31]. However, exosomes derived from other non-classical sources are recently being increasingly explored for their better accessibility, availability, as well as higher therapeutic potential. Here, we highlight important findings from recent in vivo studies with exosomes derived from synovial MSCs, infrapatellar fat pad MSCs, PRP, synovial membrane, and more, as summarized in Table 1.

#### 3.2.1. Synovial Derived Exosomes

Given that the knee joint is lined by a thin layer of synovial membrane, the function and downstream effects of EVs produced by the synovial membrane on nearby structures have been of keen interest. Several recent in vivo studies have shown promising results, with all having a relatively similar methodology, as listed in Table 1. OA was induced in all mice models via transection of the medial collateral ligament (MCL), medial meniscus, and anterior collateral ligament (ACL). All three studies exhibited the beneficial effects of SMSCs-Exos in OA models.

One of the first studies with SMSCs-exos was by Tao et al., where exosomes were taken from SMSCs with over-expressed miR-140-5p (SMSC-140-Exos), a miRNA that was shown to be involved in chondrogenic differentiation of MSCs, cartilage homeostasis, and cartilage development [20,32,33,34]. In the OA rats treated with SMSC-140-Exos, there was no significant difference in OARSI score and chondrocyte count compared to the normal healthy rats, highlighting the ability of SMSC-140-Exos to prevent severe articular cartilage damage in the knee [20].

Likewise, in a more recent study by Wang et al., EVs from SMSCs were packaged with microRNA (miR)-31 (SMSC-EV-miR-31), which has been shown to promote the growth and migration of chondrocytes [22,35]. As the authors had hypothesized, the OARSI score, degree of cartilage destruction seen on tissue sections, and levels of IL-1β, IL-6, and TNF-α were significantly decreased in OA mice treated with SMSC-EV-miR-31 compared to OA mice without treatment [22]. Therefore, SMSC-EVs-miR-31 may reduce knee articular cartilage damage and joint inflammation, similar to SMSC-140-Exos [22].

Additionally, another recent study explored the effects of modified exosomes derived from SMSCs [21]. In this study by Wang et al., exosomes were derived from miR-155-5p-overexpressing SMSCs (SMSC-155-5p-Exos), a miRNA previously shown to play a role in cell proliferation and apoptosis and one of the most downregulated in OA patients [21,36,37]. The investigators found that OA mice treated with SMSC-155-5p-Exos had a lower OARSI score, lower caspase-3 expression, higher number of chondrocytes, and higher collagen 2 (CoII) expression compared to OA mice without any treatment. Importantly, SMSC-Exos without miR-155-5p were also still able to have a significantly lower OARSI score, lower caspase-3 expression, and higher number of chondrocytes compared to OA mice without any treatment. This highlights the potential for both modified and non-modified SMSC-Exos to improve articular cartilage damage and promote cartilage regeneration in human OA patients.

#### 3.2.2. Infra-patellar Fat Pad Derived Exosomes

IPFP could serve as useful source of regenerative cells for cartilae repair as evidence shows it is responsible for some spontaneous joint repair in OA [38], and MSCs from the IPFP possess significnt chondrogenic potential [39,40]. Both MSCs and EVs including exosomes that are derived from IPFPs have been of interest for OA patients. Early success with IPFP-derived MSCs, as reviewed by Huri et al., spurred interest in IPFP-MSC derived exosomes (IPFP-Exos) [41]. Thus far, there has been only one in vivo study with IPFP-Exos for studying knee OA by Wu et al. [19].

For this first in vivo study with IPFP-Exos, as highlighted in Table 1, destabilization of the medial meniscus (DMM) surgery was used for the OA model with injections of 10 μL IPFP-Exos (10^10^ particles/mL) for 4 weeks or 6 weeks (twice a week) [19]. Mice treated with IPFP-Exos showed a significantly lower OARSI score, decreased expression of ADAMTS5 and MMP13, and increased expression of collagen II compared to the OA mice treated with the negative control (PBS only) [19]. Additionally, OA mice that received IPFP-Exos showed a duty cycle ratio of right hind foot to left hind foot closer to 1 compared to OA mice treated without IPFP-Exos [19]. Collectively, this indicates the potential for IPFP-Exos to not only reduce articular cartilage destruction but also improve gait function in DMM-induced OA mice models [19]. Further in vivo studies showed that this cartilage protection from IPFP-Exos is at least partially attributed to miR-100-5p-mediated downregulation of mammalian target of rapamycin (mTOR) and its downstream pathway in chondrocytes [19]. Additionally, though not studied in vivo, in vitro studies suggest that IPFP-Exos may also promote chondrogenesis in periosteal cells via increased expression of miR-145 and miR-221 and suppress proinflammatory cytokines, further strengthening the potential therapeutic role of IPFP-Exos in OA [42].

These cartilage protective findings of IPFP-Exos in DMM-induced OA are similar to the beneficial effects seen in a new study with ADSCs-Exos for chronic rotator cuff tears in a rabbit model, where lower fatty infiltration, more newly regenerated fibrocartilage, and improved biomechanics was seen in the ADSCs-Exos group compared to those with saline only [43].

#### 3.2.3. PRP Derived Exosomes

Platelet-rich plasma (PRP) is human plasma that has an increased concentration of platelets. PRP has been of high interest due to platelets carrying large amounts of growth factors and anti-inflammatory cytokines that promote cartilage extracellular matrix sythesis and directly inhibit the expression of MMPs and other known mediators of OA [44]. However, due to the large variation in bioactive components, PRP efficacy as an OA treatment is heterogenous. For example, PRP was found in recent meta-analyses to be overall effective in releaving knee OA related pain, but in several studies the results are not statistically significant or clinically meaningful [45,46,47]. The concentrations of bioactive proteins vary largely depending on numerous factors, such as platelet, volume of PRP, use of activating factors, and residual red and white blood cell count. Thus, this variation can be eliminated by isolating and applying only the bioactive proteins, which are released via the secretion of exosomess. Prior research suggests that exosomes at least partially mediate the effects of PRP in certain processes such as for reepithelialization of chronic cutaneous wounds and preventing GC-induced apoptosis in a rat model of osteonecrosis of the femoral head, but the exact mechanisms still remain unclear, especially in knee OA [48,49,50].

A recent in vivo study by Liu et al. further characterized these mechanisms [16]. The MCL, ACL, and medial meniscus in rabbits were transected for the OA model, with the experimental group receiving 100 μg/mL exosomes derived from PRP (PRP-Exos) by intra-articular injection once a week for 6 weeks [16]. OA rabbits treated with PRP-Exos had a significantly lower OARSI score and higher chondrocyte count compared to those without PRP-Exos [16]. Additionally, PRP-Exos in vivo promoted cartilage repair as evidenced by its reversal of the decrease in collagen II and RUNX2 expression seen in OA [16]. Subsequent in vitro studies indicated that this protection against OA is likely due to PRP-Exos role in modulating the Wnt/β-catenin signaling pathway [16].

#### 3.2.4. Amniotic Fluid Stem Cell (AFSC)-Derived Exosomes

Amniotic fluid stem cells (AFSCs) and their derived exosomes have been of particular interest ever since an early landmark study that showed for the first time the ability for AFSC-derived EVs to play a role in mediating regenerative, anti-inflammatory, and proliferation effects of stem cells in vitro and in vivo [51]. Additionally, AFSCs have the added benefit of being easily obtainable from amniotic fluid via a minimally invasive approach in the clinic, making future clinical translation readily feasible. Only one study with human AFSC-derived exosomes (AFSC-Exos) for OA has been done in vivo [17]. Zavatti et al. created a rat OA model using monoiodoacetate (MIA) injections and then treated these mice 3 weeks after with 100 ug exosomes (on the basis that this is the amount of exosomes produced by 500,000 AFSC cells) [17]. OA mice treated with AFSC-Exos showed a significantly lower OARSI score, greater percentage of cartilage covering the joint surface on histology, less fibrous-connective tissue covering the joint surface, and greater pain tolerance compared to MIA-induced OA mice that were not treated with AFSC-Exos [17]. The mechanism behind these findings is likely due to exosomes promoting M2 anti-inflammatory macrophages via TGF-β, although additional studies are needed [17].

#### 3.2.5. Umbilical cord MSCs-Derived Exosomes

Umbilical cord MSC-derived exosomes (UMSC-Exos) have also been of high interest, especially due to a recent study showing that UMSCs are able to generate four times more exosomes per cell than BMSCs or AMSCs [52]. To our knowledge, three in vivo studies with UMSC-Exos for knee cartilage defects has been conducted to date [23,24,25].

Yan and Wu created a rabbit cartilage defect model via surgery in the trochlear grooves of the distal femur and then treated them with UMSC-Exos weekly at a concentration of 1 × 10^10^/mL for four weeks [25]. UMSC-Exos produced from a 3D culture had the lowest Wakitani histological scores, highest International Cartilage Repair Society (ICRS) macroscopic assessment scores, and greater cartilage thickness with more surface regularity compared to those rabbits treated with 2D-culture UMSC-Exos or no exosomes [25]. Further in vitro studies indicate that this cartilage defect repair from 3D-UMSC-Exos is likely due to their ability to promote migration, inhibition apoptosis, and promote proliferation of chondrocytes via activation of TGF-β1 and smad2/3 signaling pathways [25].

Further studies by a similar team of researchers aimed to better elucide the mechanisms underlying those therapeuetic effects of UMSC-Exos [23,24]. First, it was demonstrated that mechanical stimulation using a rotary cell culture system could increase the expression of long non-coding RNA (lncRNA) H19 in UMSC-Exos and that this lncRNA H19 may play a role in the improved repair of cartilage defects [23]. A follow-up study was then conducted to better understand the involvement of lncRNA H19 in cartilage repair [24]. Using a cartilage defect model in rodents, the investigators showed that exosomes over-expressing lncRNA H19 (H19-Exos) lead to significantly better T2 mapping scores from MRI images and ICRS scores from macroscopic images relative to controls. Addtionally, on histological examination, H19-Exos treated rats had relatively more glycosaminoglycan deposition and collagen I and II staining. Importantly, these beneficial effects of H19-Exos were negated partially or completely when the exosomes were administered with miR-29b-3p agomir. Thus, the authors suggest that the therapeutic effect of UMSC-Exos in cartilage repair may at least partly involve the lncRNA H19/miR-29b-3p/FoxO3 axis, which is further described in their study.

#### 3.2.6. Embryonic Stem Cell-induced MSCs Derived Exosomes

Many advancements in understanding the role of exosomes derived from human embryonic mesenchymal stem cells (human eMSCs-Exos) in OA of the knee have been made, as several in vivo studies have been completed successfully with comparisons between their methodology listed in Table 1. One of the first in vivo studies using human eMSCs-Exos for OA was by Zhang et al., where osteochondral defects were created surgically in the trochlear groove of the distal femur in a rat model, following the surgical procedure, 100 ug exosomes were administered weekly for 12 weeks [11]. At the end of 12 weeks, human eMSCs-Exos treated rats had better hyaline cartilage formation and subchondral bone regeneration with good surface regularity compared to control, defect knees treated with PBS only, which primarily continued to show fibrous and non-cartilaginous tissue [11]. The same research team lead by Zhang et al. then repeated the study using the same animal model and exosome concentration but this time aimed to explore the mechanisms underlying these findings [26]. Animals treated with human eMSCs-Exos showed significant greater type 2 collagen deposition, lower type 1 collagen deposition, and lower Wakitani scores as early as 2 weeks after starting treatment compared to vehicle control rats [26]. More importantly, an increased amount of proliferative cell nuclear antigen (PCNA+) cells, increased M2 macrophages, and decreased M1 macrophages were seen at weeks 2, 6, and 12, with significantly less cleaved caspase-3 (CCP3+) apoptotic cells at week 6 compared to vehicle control rats, indicating human eMSCs-Exos are likely contributing to cartilage repair via regulating apoptosis and cellular proliferation genes [26]. Further in vitro studies showed that this regulation of apoptosis, chondrocyte proliferation, and chondrocyte migration by human eMSCs-Exos is likely mediated via AKT and ERK signaling pathways [26]. More specifically, a recent study showed that miR-135b in MSC-exosomes may downregulate Sp1 to promote chondrocyte proliferation [53], which may explain the results seen in the studies by Zhang et al. [11,26]. Consistent with these studies by Zhang et al., Wang et al. showed in a DMM model for OA that human eMSCs-Exos may play a role in cartilage repair by increasing collagen type 2 synthesis and decreasing ADAMTS5 expression in presence of Il-1β, thereby modulating cartilage extracellular matrix synthesis and degradation [18].

## 4. Conclusions and Future Directions

The inability of intrinsic MSCs to completely reverse or inhibit the progression of OA highlights the need for therapies based on exogenous exosomes derived from MSCs or other tissues. Early research for both classic and non-classic sources of exosomes has shown promising results for the prevention and treatment of knee OA. However, prior to clinical translation, a further understanding of many different aspects of exosomes is needed. One area in particular is the mechanisms underlying the therapeutic effects of exosomes. For example, it is possible that the exosomes derived from synovium tissue have an improved cartilage regenerative capacity relative to exosomes from other tissue sources due to their closer vicinity to the joint cartilage and phenotypic features. As such, there has been interest in using MSCs directly from synovial fluid (SF-MSCs). Although no in vivo studies using exosomes derived SF-MSCs have been done yet, several studies have reported the chondrogenic potential of SF-MSCs [54,55]. The origin of these cells is not fully elucidated but in vivo studies suggest that they may derive from synovial membranes [56]. Synovial fluid offers several advantages over the classical sources as well as the synovial membrane: (1) SF-MSCs can be harvested rapidly during a routine arthroscopy, eliminating the need for an invasive operation and anesthesia, (2) compared to bone marrow MSCs, SF-MSCs express increased hyaluronan receptor CD44 and uridine diphosphoglucose dehydrogenase (UDPGD), an enzyme required for hyaluronan synthesis [57]. Li et al. demonstrated superior chondrogenic and reparative potential of human SF-MSC in vitro and in vivo [58]. Similar findings were obtained by Zayed et al. in a rat cartilage defect model, where transplantation of xeogenic SF-MSC resulted in better healing and significantly higher collagen-II synthesis [59]. These promising results with SF-MSCs highlights the need for future studies to focus on the effects of their exosomes on in vivo models of knee OA.

Given the success highlighted in this review with exosomes from various non-classic MSC tissue sources, it is essential to continue exploring the therapeutic potential of exosomes derived other tissue sources. For example, two sources of interest include ligament-derived exosomes and subchondral bone-derived exosomes. Prior research to our knowledge has only included in vitro studies of exosomes derived from these sources and thus was not extensively reviewed here. Briefly, exosome-like EVs can be produced from osteoblast cells derived from subchondral bone in OA patients, which may then play a role in subchondral bone remodeling via modulation of TGF-B signaling, though this is unclear and needs further exploration [60,61]. Additionally, in regards to ligament-derived exosomes, periodontal ligament-derived exosomes have been shown to play a role in modulating periodontal inflammatory homeostasis via inhibition of the NF-κB signaling pathway [62,63]. Furthermore, exosome-educated macrophages (EEMs) and MSC-derived exosomes (MSC-Exos) have recently been shown to improve ligament healing in a rat medial collateral ligament (MCL) injury model, with MSC-Exos reducing scar formation and increasing type 1 and type 3 collagen production [64]. Though not derived from ligaments at the knee joint, these early results are promising and necessitate the need to investigate exosomes derived from the anterior cruciate ligament, MCL, other knee ligaments, and other body tissues. As such, our understanding of both ligament-derived exosomes and subchondral-bone derived exosomes is in its infancy and requires further studies with eventual in vivo studies as well.

More importantly, our general understanding of the biological, functional, and physical properties of exosomes needs to be further developed. One such area that requires a deeper examination includes exosome isolation methods. For example, ultracentrifugation is currently the most commonly used method of exosome isolation in studies. However, ultracentrifugation is not readily scalable for mass production and it can promote vesicle aggregation and co-isolation of soluble proteins, possibly influencing the results seen in human clinical studies [65]. Various exosome isolation techniques from multiple starting materials have been extensively studied and reviewed [66,67,68], and the consensus seems to have shifted towards more “gentle” isolation techniques that could improve exosome purity and integrity. One example is size exclusion chromatography (SEC), which has been shown to effectively separate exosomes from proteins and soluble factors in different biological fluids [69,70].

Furthermore, when examining the preclinical outcomes of the exosomes, more attention needs to be paid to the choice of animal models. For example, in one of the studies reviewed in this article, the MIA induced model was used, which, despite its suitability in studying drug effects on inflammation and pain induced by the substance, does not translate very well to the more complexed osteoarthritis development and pathology in the human knee [71,72]. More importantly, although small animals such as mice, rats, and rabbits are advantageous for the initial line of investigation, studies using small animals have limited translatability due to differences in intrinsic repair capabilities and histological properties compared to human knee cartilage. Therefore, future studies using larger animals such as pigs or goats are needed [73].

Another area that needs to be explored is the influence and efficacy of the delivery system used for the exosomes. In most, if not all, of the OA animal models, the treatment regimen was usually completed within a matter of weeks with only a weekly injection of exosomes. However, the larger size of the human knee joint and complexity of OA in humans will require significantly higher does and longer periods of treatment. This creates a need to improve the retention of exosomes in the joint cavity. By immobilizing exosomes in a suitable bio-compatible material such as hydrogels or scaffolds, it is possible to establish a controlled release of and extend the half-life of injected exosomes, thereby reducing the necessary dosage and administration frequency. An increasing number of studies have examined different ways of encapsulating exosomes to extend the retention of exosomes, and these materials and methods have been well reviewed by Riau et al. [74]. Because exosomes are very similar in terms of physical structure and chemical composition [75,76], delivery methods in viral drugs may lend some insights on how to ultimately achieve sustained release of exosomes in various tissues.

Moving forward, many aspects of exosome research still need to be thoroughly explored, but in particular is the contents of exosomes, including any proteins, RNAs, small RNAs, miRNAs, and more. As mentioned in several studies earlier, many of the downstream therapeutic effects of exosomes are likely due to these contents targeting the expression of certain genes at the target cell. For example, as described earlier, the role of SMSCs with over-expressed miR-140-5p in prevention of OA in vivo is likely due to miR-140-5p suppressing RalA expression to rescue Sox9 expression, which then promotes ECM secretion [20]. In an experimental OA model, miR-100-5p was responsible for the protective effects of IPFP-Exos by stimulating anabolic and inhibiting catabolic activities of chondrocytes and suppressing inflammation via the mTOR signaling pathway [19]. A more comprehensive summary of the potential roles of important exosomal miRNAs found in MSCs and chondrocytes can be found in a recent review by Ni et al. [60]. Further investigation into these contents of exosomes and their effects on not only chondrocytes but other surrounding intra- and extra-articular cell types is needed. Moreover, understanding if contents vary in different exosomes when derived from different sources is needed. Additionally, analysis into the stability of these contents for isolation and storage in the future is needed. Collectively, such understanding may allow us in the future to incorporate certain contents into exosomes for clinical translation.

In moving towards clinical use, many challenges in using exosomes still exist, including but not limited to dosage, frequency, safety, kinetics, and control batch differences. For exosomes derived from different sources, finding the ideal concentrations and optimal frequency of exosome administration in animal models and humans needed for therapeutic effects is critical. Likewise, standardization of the isolation, extraction, and storage processes is needed as well. In regards to safety, an early study has shown transplantation of human UMSC-Exos to be safe in animal models [77], and others have reported minimal immunogenicity of MSC-derived exosomes [78,79]. Nevertheless, study of immunogenicity of exosomes from an allogenic source might be necessary. More similar studies using the aforementioned exosomal sources is needed. As much of our current research and understanding is in the early stage, there remains much time until regulatory guidelines for exosomes can be properly established by the Food and Drug Administration (FDA). Some of the key considerations regarding the future directions of exosomes as therapies are summarized in Figure 2.

## Figures and Tables

**Figure 1 jcm-10-02001-f001:**
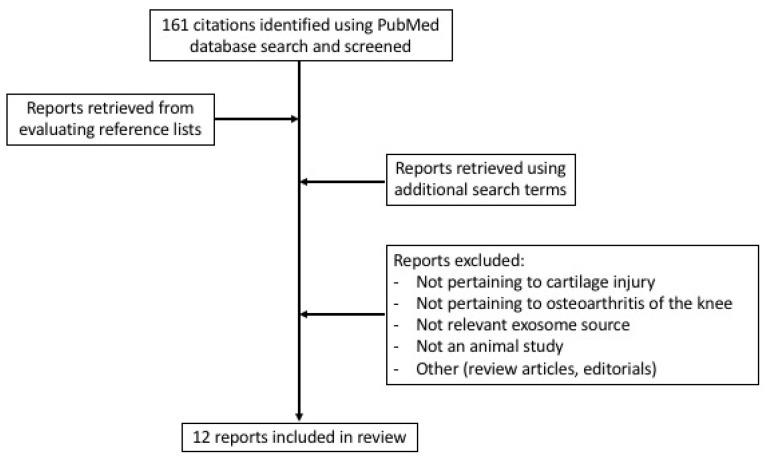
Flow diagram of literature search conducted including criteria and selection.

**Figure 2 jcm-10-02001-f002:**
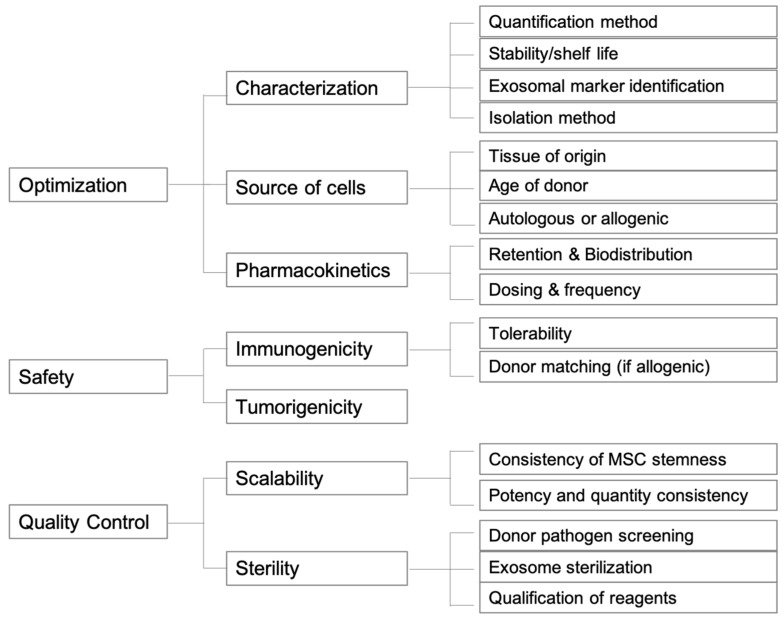
Graphical scheme of future directions for exosome therapeutic products.

**Table 1 jcm-10-02001-t001:** In vivo studies with exosomes derived from non-classic sources for knee osteoarthritis and cartilage injuries.

Source	Dose/Volume	Animal	Animal Model	Results	Reference
Platelet-Rich Plasma	100 μg/mL	Rabbits	Transection of MCL, medial meniscus, ACL	Reversed the decrease in collagen II and RUNX2 protein expression, promoted cartilage repair, inhibited OA	Liu et al., 2019 [16]
Infrapatellar fat pad	10 μL 10^10^ particles/mL	Mice	Destabilization of the medial meniscus (DMM) surgery	Alleviate articular cartilage damage and improve gait, likely via miR-100-5p downregulation of mTOR	Wu et al., 2019 [17]
Synovial Membrane	100 μL 10^11^ particles/mL	Rats	Transection of MCL, medial meniscus, ACL	Enhance cartilage tissue regeneration and prevent OA	Tao et al., 2017 [18]
Synovial Membrane	5 μL particles/mL	Mice	Transection of MCL, medial meniscus, ACL	Reduced cartilage damage and restored structure of cartilage surface	Wang et al., 2020 [20]
Synovial Membrane	30 μL 10^11^ particles/mL	Mice	Transection of MCL, medial meniscus, ACL	Prevent OA, promote cartilage regeneration and improve articular cartilage damage	Wang et al., 2020 [19]
Umbilical cord	500 μL 10^10^ particles/mL	Rabbits	rabbit cartilage defect model via surgery at the trochlear grooves of the distal femur	Repair cartilage defects via promoting migration and proliferation of chondrocytes	Yan and Wu, 2019 [23]
Umbilical Cord	100 μL 1 mg/mL	Rats	Distal femur cartilage defect	Improved cartilage defects via increased collagen II secretion and matrix synthesis, possibly involving lncRNA H19	Yan et al., 2020 [21]
Umbilical Cord	200 μl injection 1 mg/mL	Rats	Unilateral cartilage defect on the femoral trochelear groove	Improve cartilage repair via lncRNA H19/miR-29b-3p/FoxO3 axis	Yan et al., 2021 [22]
Human Embryonic Stem Cell	100 µg exosomes per 100 μL injection	Rats	Osteochondral defects surgically created at trochlear grove of distal femur	Better hyaline cartilage formation and subchondral bone regeneration	Zhang et al., 2016 [11]
Human Embryonic Stem Cell	100 µg exosomes per 100 μL injection	Rats	Osteochondral defects surgically created at trochlear grove of distal femur	Regulate apoptotic and cellular proliferation genes	Zhang et al., 2018 [24]
Human Embryonic Stem Cell	5 µL exosomes	Mice	Destabilization of the medial meniscus (DMM) surgery	Modulate cartilage extracellular matrix synthesis and degradation	Wang et al., 2017 [26]
Amniotic Fluid	100 µg exosomes in 50 µL	Rats	Monoiodoacetate (MIA) injections	Protection from cartilage damage	Zavatti et al., 2019 [25]
Tendon, Ligament, Subchondral	-	-	-	-	-

Abbreviations: MCL, medial collateral ligament; ACL, anterior cruciate ligament.

## Data Availability

Not applicable.

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
