# Peer review of "Exosomes Derived from Non-Classic Sources for Treatment of Post-Traumatic Osteoarthritis and Cartilage Injury of the Knee: In Vivo Review"

_jcm, 2021, doi:10.3390/jcm10092001_

Round 1

Reviewer 1 Report

The authors performed a review of in vivo studies using exosomes (from sources other than bone marrow or adipose tissue) in OA animal models. The review is interesting and provides some contribution to the field.

I have the following minor comments.

1) In the last paragraph of the Introduction it is said that the use of exosomes from classical sources for knee OA has been extensively reviewed elsewhere. However, there are no references to this previous work.

2) Table 1 should alway include the animal species in which the studies have been conducted: although this is mainly mice, there may be some rat study and in one case rabbits are appropriately mentioned. This should avoid confusion.

3) It is unclear why tendon-derived exosomes are included in the main text and not only in the Discussion as it was done for subchondral bone-derived material. Although I have nothing against it, this discordance should be appropriately handled. If tendon-derived exosomes remain in the main text, they should be deal with as the last source, since there are no OA-specific in vivo studies. 

4) I would be careful in stating that PRP has "demonstrated efficacy" in OA, as this is highly debated in current treatment guidelines.

5) In the amniotic fluid-derived exosomes review, the authors take into account the rat model using MIA. This is fine, but it should be made very clear that this model is different than surgical OA models.

6) In the penultimate  paragraph of the Discussion, the authors make reference to a table in ref.50. This is a bit weird. If it not possible to reproduce the table with permission, the teble itself should be better described.

Author Response

We thank the reviewers for their insightful comments that will help improve our manuscript. We agree with their comments and are making corresponding changes in the manuscript.

In response to the comments:

  1. In the last paragraph of the Introduction it is said that the use of exosomes from classical sources for knee OA has been extensively reviewed elsewhere. However, there are no references to this previous work.

Those references are now included.

  1. Table 1 should always include the animal species in which the studies have been conducted: although this is mainly mice, there may be some rat study and in one case rabbits are appropriately mentioned. This should avoid confusion.

Species are now added.

  1. It is unclear why tendon-derived exosomes are included in the main text and not only in the Discussion as it was done for subchondral bone-derived material. Although I have nothing against it, this discordance should be appropriately handled. If tendon-derived exosomes remain in the main text, they should be deal with as the last source, since there are no OA-specific in vivo studies. 

Tendon-derived exosomes were removed from the Results section and are discussed as one of the potential sources for exosomes in Discussion section.

  1. I would be careful in stating that PRP has "demonstrated efficacy" in OA, as this is highly debated in current treatment guidelines.

This statement was corrected and the reason for the discrepancy was briefly addressed.

  1. In the amniotic fluid-derived exosomes review, the authors take into account the rat model using MIA. This is fine, but it should be made very clear that this model is different than surgical OA models.

The validity of this model is now discussed.

  1. In the penultimate paragraph of the Discussion, the authors make reference to a table in ref.50. This is a bit weird. If it not possible to reproduce the table with permission, the table itself should be better described.

The information is now described instead.

Thanks again for your review and look forward to your final decision on our manuscript.

Reviewer 2 Report

The authors provided a review regarding the role of exosomes in knee OA, focusing on especially in vivo studies. Though the review reads well and is nicely presented, this reviewer doubts its additional value to thew existing literature.  For instance, recently a similar review has been published (Tan SSH, Tjio CKE, Wong JRY, Wong KL, Chew JRJ, Hui JHP, Toh WS. Mesenchymal Stem Cell Exosomes for Cartilage Regeneration: A Systematic Review of Preclinical In Vivo Studies. Tissue Eng Part B Rev. 2021 Feb;27(1):1-13. doi: 10.1089/ten.TEB.2019.0326. Epub 2020 Apr 15. PMID: 32159464.), which has systematic approach used. What makes this review different from the one recently published?

Moreover, the authors make several assumptions as given though in my opinion there is still no conclusive evidence or scientific consensus. For example, the authors state in line 41-43 that MSCs have been shown both in in vitro and in vivo to play an important role in OA. On itself I can agree with this statement, however, proper clinical evidence through well executed controlled randomized trials haven’t been provided yet. Another example, is the role PRP. This is also a type treatment which is not proven in well executed (clinical studies).

Though this is a descriptive review it would be helpful to have a search strategy and inclusion/exclusion to have a better understanding for the choices made to include in the paper. Especially when studied as discussed which make use of models which are not representative for osteoarthritis or have poor translational value (MIA model, osteochondral effects). Also, only positive effects are mentioned, no studies are mentioned which show opposite effects. As such the paper feels a bit unbalanced and lacks nuance.

It would be helpful to add in table 1 the species of animals used.

Author Response

We thank the reviewers for their insightful comments that will help improve our manuscript. We agree with their comments and are making corresponding changes in the manuscript.

In response to the comments:

1. PRP efficacy is debatable 

This statement about PRP was corrected and the reason for the discrepancy was briefly addressed.

2. Species of animals should be included in the table 

Species are now added.

3. MIA model has poor translational value 

The validity of this model is now discussed.

4. Though this is a descriptive review it would be helpful to have a search strategy and inclusion/exclusion to have a better understanding for the choices made to include in the paper.

A Method section and figure illustrating the workflow have been added.

5. Also, only positive effects are mentioned, no studies are mentioned which show opposite effects. As such the paper feels a bit unbalanced and lacks nuance.

We agree, but no studies were available for searching with negative results.

However, we do disagree with the rest of the comments:

  1. Though the review reads well and is nicely presented, this reviewer doubts its additional value to thew existing literature.  For instance, recently a similar review has been published (Tan SSH, Tjio CKE, Wong JRY, Wong KL, Chew JRJ, Hui JHP, Toh WS. Mesenchymal Stem Cell Exosomes for Cartilage Regeneration: A Systematic Review of Preclinical In Vivo Studies. Tissue Eng Part B Rev. 2021 Feb;27(1):1-13. doi: 10.1089/ten.TEB.2019.0326. Epub 2020 Apr 15. PMID: 32159464.), which has systematic approach used. What makes this review different from the one recently published?”

Firstly, we believed our review provides useful value in addition to the existing literature. This is a focused review giving intensive attention to the studies that seek to use non-classical/alternative sources for exosome preparation that is intended for in vivo work. Because MSCs isolated from these non-classical sources have demonstrated equal or superior regenerative effects than the bone marrow or adipose derived MSCs and the isolation is less invasive, the use of the non-classical sources could open a new avenue for exosomes as an orthobiologic reagent.

Second, in the review mentioned by the reviewer, even though part of the non-classical sources were summarized, specifically, all three studies using human embryonic stem cell as source were reviewed, which is also what we did, but only two studies using synovial membrane as source for exosome and one study using infrapatellar fat pad as source for exosome were reviewed, which is incomplete. Furthermore, the umbilical cord-, amniotic fluid-, and the platelet-rich-plasma- derived MSC as sources for exosome preparation were not reviewed.

Therefore, our review provides an exhaustive reference for non-classical MSCs sources, as opposed to the classical sources which include bone marrow or adipose tissue.  

  1. “Moreover, the authors make several assumptions as given though in my opinion there is still no conclusive evidence or scientific consensus. For example, the authors state in line 41-43 that MSCs have been shown both in in vitro and in vivo to play an important role in OA. On itself I can agree with this statement, however, proper clinical evidence through well executed controlled randomized trials haven’t been provided yet.”

The statement that “MSCs have been shown both in vivo and in vitro to play an important role in protecting against the development of OA or articular cartilage destruction via anti-apoptotic activity, anti-fibrotic activity, promoting chondrogenesis, immunosuppression, regulating metabolic activity, and more” is not an assumption, but a conclusion from the findings from the in vitro and in vivo studies published to date. We did not and could not claim its clinical efficacy because there is not enough clinical evidence as to whether MSC-derived exosomes would work as well as they do in pre-clinical animal models. But it is a proven fact, at least in animal studies, that these exosomes “play an important role in protecting against the development of OA or articular cartilage destruction”.

Thanks again for your review and look forward to your final decision on our manuscript.

Round 2

Reviewer 2 Report

The authors have improved there manuscript and tried to convince this reviewer that this review has additional value. Based on their comments I understand (and appreciate) their effort. With respect to the discussion regarding the role of MSCs in protecting against OA I agree that the evidence provided in the preclinical work indicate this, however, in the real world human disease this is much more complicated. Intrinsic MSCs are apparently not capable to overcome the progression of OA and so far, as stated previously, no good human clinical studies have supported the preclinical findings. But this is a bit of a semantic discussion, as I understand the point of view of the authors.

Author Response

We thank the reviewer again for his/her valuable comments, and we have accommodated the comment in lines 284-285.